# A Variational Perspective on Diffusion-Based Generative Models and Score Matching

**Chin-Wei Huang** [1]  **Jae Hyun Lim** [1]  **Aaron Courville** [1]

## Abstract

Discrete-time diffusion-based generative models and score matching methods have shown promising results in modeling high-dimensional image data. Recently, Song et al. (2021) show that diffusion processes can be reverted via learning the score function, *i.e.* the gradient of the log-density of the perturbed data. They propose to plug the learned score function into an inverse formula to define a generative diffusion process. Despite the empirical success, a theoretical underpinning of this procedure is still lacking. In this work, we approach the (continuous-time) generative diffusion directly and derive a variational framework for likelihood estimation, which includes continuous-time normalizing flows as a special case, and can be seen as an infinitely deep variational autoencoder. Under this framework, we show that minimizing the score-matching loss is equivalent to maximizing the ELBO of the plug-in reverse SDE proposed by Song et al. (2021), bridging the theoretical gap.

## 1. Introduction

Generative modeling can be thought of as inverting an inference process. If the inference process is invertible, then one can focus on transforming the data into a tractable distribution (Dinh et al., 2016). If the inference process is deterministic yet non-invertible, one could learn to invert it stochastically (Dinh et al., 2019; Nielsen et al., 2020). Most generally, both inference and generation can be stochastic. This is known as the variational autoencoder (Kingma & Welling, 2014; Rezende et al., 2014, VAE).

Under the variational framework, one has a lot of flexibility in choosing the generative and inference models. Recent

---

[1]Mila, University of Montreal. Correspondence to: Chin-Wei Huang <cw.huang427@gmail.com>.

Third workshop on *Invertible Neural Networks, Normalizing Flows, and Explicit Likelihood Models* (ICML 2021). Copyright 2021 by the author(s).

work on diffusion-based modeling (Sohl-Dickstein et al., 2015; Ho et al., 2020) can be thought of as removing one degree of freedom, by freezing the inference path. The inference model is a fixed discrete-time Markov chain, that slowly transforms the data into a tractable prior, such as the standard normal distribution. The generative model is another Markov chain that is trained to revert this process iteratively. Diffusion-based models have been shown to perform remarkably well on image synthesis (Dhariwal & Nichol, 2021), rivaling the performance of state-of-the-art Generative Adversarial Networks (Brock et al., 2018).

Song et al. (2021) connect diffusion-based model and *score matching* (Hyvärinen & Dayan, 2005), by looking at the stochastic differential equation (SDE) associated with the inference process. They realize that the dynamic of the inference process can be inverted if one has access to the score function of the perturbed data, by solving another SDE reversed in time. They then propose to learn the score function of the inference process and substitute the approximate score into the formula of the reverse SDE to obtain a generative model. We call the resulting generative model the plug-in reverse SDE.

Conceptually simple as this learning procedure may seem, little is known about how the score matching loss relates to the plug-in reverse SDE. In this paper, we propose a variational framework suitable for likelihood estimation for general generative diffusion processes, and use this framework to connect score matching with maximum likelihood. We do so by combining two important theorems in stochastic calculus: the *Feynman-Kac formula* for representing the marginal density of the generative diffusion as an expectation (Section 3), and the *Girsanov theorem* for performing inference in function space (Section 4). By reparameterizing our generative and inference SDEs, we obtain a training objective equivalent to minimizing the score matching loss (Section 5). Our theory suggests that by matching the score, one actually maximizes a lower bound on the log marginal density of the plug-in reverse SDE, laying a theoretical foundation for this learning procedure.

**Notation:** We use $(Y_s, s)$ to denote the inference process (where $Y_0$ is the data), and $(X_t, t)$ to denote the generative

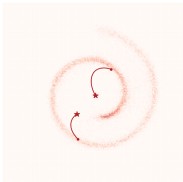 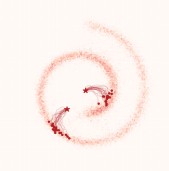 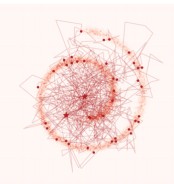

*Figure 1.* Three special cases of generative SDEs. The stars indicate the initial values, followed by some random sample paths. *Left*: trained with no diffusion $\sigma = 0$ (*i.e.* neural ODE). *Middle*: trained with some fixed diffusion $\sigma > 0$. *Right*: trained with a fixed inference process, $f$ and $g$ (*i.e.* the plug-in reverse SDE).

process (where $X_0$ is a random variable following an unstructured prior). We use $s$ and $t$ to distinguish the two directions, and always integrate the differential equations from 0 to $T > 0$ (different from the literature, where sometimes one might see integration from $T$ to 0). $\hat{B}_s$ and $B_t$ denote the Brownian motions associated with the inference and generative SDEs, respectively. $B_s'$ is a reparameterization of $\hat{B}_s$ (see Section 4). $q(y, s)$ and $p(x, t)$ denote the probability density functions of $Y_s$ and $X_t$, respectively. We let $\boldsymbol{s}_\theta$ denote a time-indexed parameterized function that will be used to approximate the score $\nabla \log q(y, s)$. $\nabla$ is the gradient wrt the spatial variable ($x$ or $y$, which we sometimes call position), $\partial_t$, $\partial_s$ and $\partial_{x_i}$ are partial derivatives, and $H_*$ denotes Hessian.

## 2. Background

Assume $Y_0$ follows the data distribution $q(y, 0)$, and $Y_s$ satisfies the Itô SDE (Øksendal, 2003)

$$\mathrm{d}Y = f \, \mathrm{d}s + g \, \mathrm{d}\hat{B}_s \qquad (1)$$

where $f$ and $g$ are chosen such that the density $q(y, s)$ will converge to some tractable prior $p_0$ as $s \to T$. It is possible to find a "reverse" SDE, whose marginal density evolves according to $q(y, s)$, reversed in time. Assume $g(y, t) = g(t)$. An example[1] used in Song et al. (2021) is

$$\mathrm{d}X = (gg^\top \nabla \log q(X, T - t) - f) \, \mathrm{d}t + g \, \mathrm{d}B_t. \qquad (2)$$

If $X_0 \sim p_0$, then the density $p(x, t)$ of $X_t$ is equal to $q(x, T - t)$. This means if we have access to the score function $\nabla \log q$, we can solve the above SDE to obtain $X_T \overset{d}{=} Y_0$. Song et al. (2021) propose to approximate the score via a parameterized score function $\boldsymbol{s}_\theta$ by minimizing

$$\int_0^T \mathbb{E}_{Y_s} \left[ \frac{1}{2} ||\boldsymbol{s}_\theta(Y_s, s) - \nabla \log q(Y_s, s)||_{\Lambda(s)}^2 \right] \mathrm{d}s$$

[1]See Appendix C for a family of equivalent (reverse) SDEs indexed by some parameter $\lambda$, of which equation (2) is a special case with $\lambda = 0$.

| Method | Loss |
|---|---|
| $\mathcal{L}_{\text{ESM}}$ | $\frac{1}{2}\mathbb{E}[||\boldsymbol{s}_\theta(Y_s, s) - \nabla \log q(Y_s)||_\Lambda^2]$ |
| $\mathcal{L}_{\text{ISM}}$ | $\mathbb{E}[\frac{1}{2}||\boldsymbol{s}_\theta(Y_s, s)||_\Lambda^2 + \nabla \cdot (\Lambda^\top \boldsymbol{s}_\theta)]$ |
| $\mathcal{L}_{\text{SSM}}$ | $\mathbb{E}[\frac{1}{2}||\boldsymbol{s}_\theta(Y_s, s)||_\Lambda^2 + v^\top \nabla(\Lambda^\top \boldsymbol{s}_\theta)v]$ |
| $\mathcal{L}_{\text{DSM}}$ | $\frac{1}{2}\mathbb{E}[||\boldsymbol{s}_\theta(Y_s, s) - \nabla \log q(Y_s|Y_0)||_\Lambda^2]$ |

*Table 1.* Score matching losses. $v$ follows the Rademacher distribution.

where the expectation in the integral is known as the explicit score matching (ESM) loss $\mathcal{L}_{\text{ESM}}$, and $\Lambda(s)$ is a positive definite matrix[2]. $\mathcal{L}_{\text{ESM}}$ is not immediately useful, since we do not have access to the ground truth score $\nabla \log q$. A few alternative losses can be used, which are all equal to one another up to a constant, including implicit score matching (Hyvärinen & Dayan, 2005, ISM), sliced score matching (Song et al., 2020, SSM), and denoising score matching (Vincent, 2011, DSM). The losses (summarized in Table 1) are related through the following identity:

$$\mathcal{L}_{\text{ESM}} - \frac{1}{2}\mathcal{I}(q(y_s, s)) = \mathcal{L}_{\text{ISM}} = \mathcal{L}_{\text{SSM}}$$

$$= \mathcal{L}_{\text{DSM}} - \frac{1}{2}\mathbb{E}_{y_0}[\mathcal{I}(q(y_s, s|y_0))], \qquad (3)$$

where $\mathcal{I}(q)$ denotes the Fisher information $\mathcal{I}(q) = \mathbb{E}[||\nabla \log q||_\Lambda^2]$, which is a constant. After training, Song et al. (2021) plug $\boldsymbol{s}_\theta$ into (2) to define a generative model. We refer to this SDE as the *plug-in reverse SDE*. The plug-in reverse SDE has been demonstrated to have impressive empirical results, but a theoretical underpinning of this learning framework is still lacking. For example, it is unclear how minimizing the score matching loss relates to the sampling procedure, *e.g.* whether the probability distribution induced by the plug-in reverse SDE gets closer to the data distribution in the sense of any statistical divergence or metric. We seek to answer the following question in this paper: *How will minimizing the score-matching loss impact the plug-in reverse SDE?* We first provide a framework to estimate the likelihood of generative SDEs, and then get back to this question in Section 5.

## 3. Stochastic instantaneous change of variable

Let $X_t$ be a diffusion process solving the following Itô SDE:

$$\mathrm{d}X = \mu(X, t) \, \mathrm{d}t + \sigma(X, t) \, \mathrm{d}B_t \qquad (4)$$

with the initial condition $X_0 \sim p_0$. We use this SDE as the generative SDE. We are interested in the density function

[2]We use this matrix to induce a Mahalanobis norm $||x||_\Lambda^2 := x^\top \Lambda x$, which will be used in Section 5.

of $X_t$, which follows the *Fokker Planck equation*:

$$\partial_t p(x,t) =$$
$$-\sum_j \partial_{x_j}[\mu_j(x,t)\,p(x,t)] + \sum_{i,j} \partial^2_{x_i,x_j}[D_{ij}(x,t)\,p(x,t)] \tag{5}$$

with the initial value $p(\cdot,0) = p_0(\cdot)$, where $D = \frac{1}{2}\sigma\sigma^T$ is the diffusion matrix. We can expand the Fokker Planck and rearrange the terms to obtain

$$\partial_t p(x,t) = \left[ -\nabla \cdot \mu(x,t) + \sum_{i,j} \partial^2_{x_i,x_j} D_{ij}(x,t) \right] p(x,t) +$$
$$\sum_i \left[ -\mu_i(x,t) + 2 \sum_j \partial_{x_i} D_{ij}(x,t) \right] \partial_{x_i} p(x,t) +$$
$$\sum_{i,j} D_{ij}(x,t) \partial^2_{x_i,x_j} p(x,t) \tag{6}$$

so that all coefficients of the same order are grouped together. For simplicity, we assume $\sigma$ is independent of $x$ throughout the paper[3]. Then (6) reduces to

$$\partial_t p(x,t) = -(\nabla \cdot \mu)\,p - \mu^\top \nabla p + D : H_p \tag{7}$$

where : denotes the Frobenius inner product between matrices. Even with this simplification, solving (7) is not trivial. Fortunately, we can estimate this quantity by applying the *Feynman-Kac formula* (presented in Appendix E), which allows us to rewrite the solution of the Fokker Planck PDE using a probabilistic representation.

$$p(x,T) =$$
$$\mathbb{E}\left[ p_0(Y_T) \exp\left( \int_0^T -\nabla \cdot \mu(Y_{s'}, T - s')\,\mathrm{d}s' \right) \,\Bigg|\, Y_0 = x \right] \tag{8}$$

where $Y_s$ is a diffusion process solving

$$\mathrm{d}Y = -\mu(Y, T - s)\,\mathrm{d}s + \sigma(Y, T - s)\,\mathrm{d}B'_s, \tag{9}$$

**Remark 1 (Marginalization).** This representation can be interpreted as a mixture of continuous time flows. Assume a sample path of the Brownian motion is given, and we are interested in how the density evolves following the dynamic (4). In the infinitesimal setting, it can been seen as applying the invertible map $x \mapsto x + \mu(x,t)\Delta t + \sigma(t)\Delta B_i$, where $\Delta B_i := B_{(i+1)\Delta t} - B_{i\Delta t}$ is the Brownian increment. Since $\sigma$ is independent of the spatial variable, it can be seen as a constant additive transformation, which is volume preserving, so it will not be taken into account when

---

computing the change of density. The only contribution to the change of density will be from the log-determinant of the Jacobian of $\mathrm{id} + \mu$, which means we can simply apply the instantaneous change of variable formula (Chen et al., 2018). This will be the conditional density given the entire $\{B_t : t \geq 0\}$, and marginalizing it out results in the expectation in (8). See Appendix B for details.

## 4. Inferring latent Brownian motion

As our goal is to estimate likelihood, we would like to compute the log density value using (8). However, this involves integrating out all possible Brownian paths, which is intractable. To resolve this, we view the Brownian motion as a latent variable, and perform inference by assigning higher probability to sample paths that are more likely to generate the observation. One can view this as a VAE, except we have an infinite dimensional latent variable.

Formally, let $(\Omega, \mathcal{F}, \mathbb{P})$ be the underlying probability space for which $B'_s$ is a Brownian motion. Suppose $\mathbb{Q}$ is another probability measure on $(\Omega, \mathcal{F})$ *equivalent* to $\mathbb{P}$; that is, $\mathbb{P}$ and $\mathbb{Q}$ are similar in the sense that they have the same measure zero sets. This allows us to apply the change-of-measure trick and lower bound the log-likelihood with a finite quantity using Jensen's inequality:

$$\log p(x,T) \geq \mathbb{E}_{\mathbb{Q}}\left[ \log \tfrac{\mathrm{d}\mathbb{P}}{\mathrm{d}\mathbb{Q}} + \log p_0(Y_T) - \int_0^T \nabla \cdot \mu \,\Bigg|\, Y_0 = x \right] \tag{10}$$

Note that $\frac{\mathrm{d}\mathbb{P}}{\mathrm{d}\mathbb{Q}}$ is the *Radon-Nikodym* derivative of $\mathbb{P}$ wrt $\mathbb{Q}$. When both measures are absolutely continuous wrt a third measure, say Lebesgue, then the derivative can be expressed as the ratio of the two densities. However, since we are dealing with an infinite dimensional space, we are immediately faced with the following problems: (1) Is there a measure $\mathbb{Q}$ (equiv. to $\mathbb{P}$) for which $\frac{\mathrm{d}\mathbb{P}}{\mathrm{d}\mathbb{Q}}$ can be easily computed, or at least numerically approximated? (2) Can we find a reparameterization of $B'_s$ under the new law $\mathbb{Q}$ to estimate the gradient needed for training?

We resort to the *Girsanov theorem*, which describes a general framework for dealing with the change of measure of Gaussian random variables under additive perturbation. It allows us to consider the law of a diffusion process as $\mathbb{Q}$. Specifically, equation (29) provides a standarization formula of $B'_s$ under $\mathbb{Q}$, which means we can "invert" it to reparameterize $B'_s$.

**Theorem 1 (Continuous-time ELBO).** *Let $\mathbb{Q}$ be defined by (30). Then the RHS of (10) can be rewritten as*

$$\mathbb{E}\left[ -\tfrac{1}{2} \int_0^T \|a(\omega,s)\|_2^2\,ds + \log p_0(Y_T) - \int_0^T \nabla \cdot \mu \, ds \,\Bigg|\, Y_0 = x \right] \tag{11}$$

---

[3]Our framework also works with the most general case, but the formula needs to be adapted to account for the spatial partial derivatives. See Appendix A for the general case.

*where the expectation is taken wrt the Brownian motion $\hat{B}_s$, and $Y_s$ solves[4]*

$$dY = (-\mu + \sigma a)\, ds + \sigma d\hat{B}_s \qquad (12)$$

We call $Y_s$ solving (12) the inference SDE, and (11) the continuous-time ELBO (CT-ELBO), denoted by $\mathcal{E}^\infty$. We defer the proof and a few remarks to Appendix G. In Appendix H, we show that this ELBO is the limit of the traditional ELBO for discretized diffusion models, so maximizing it can be seen as training an infinitely deep VAE.

We visualize the density of special cases of generative SDEs trained by maximizing the lower bound in Figure 1.

## 5. Score-based generative modeling

Recall that our goal is to analyze the plug-in reverse SDE and draw connection to score matching. To this end, we reparameterize the generative and inference SDEs as

$$\mathrm{d}X = (gg^\top s_\theta - f)\, \mathrm{d}t + g\, \mathrm{d}B_t \ \text{ and } \ \mathrm{d}Y = f\, \mathrm{d}s + g\, \mathrm{d}\hat{B}_s \qquad (13)$$

by letting $a = g^\top s_\theta$, where the time variable is reversed $(T - t)$ for the generative process, and forward in time $(s)$ for inference. The ELBO (11) can be rewritten as

$$\mathcal{E}^\infty = \mathbb{E}_{Y_T}[\log p_0(Y_T)\,|\,Y_0 = x] -$$
$$\int_0^T \mathbb{E}_{Y_s}\left[\frac{1}{2}||s_\theta||_{gg^\top}^2 + \nabla \cdot (gg^\top s_\theta - f)\,\middle|\,Y_0 = x\right] ds \qquad (14)$$

Comparing the integrand to the implicit score matching loss in Table 1, we immediately see that the network $s_\theta$ approximates $\nabla \log q(y, s)$, the score function of the marginal density of $Y_s$. That is, *matching the score of $q(y, t)$ amounts to maximizing the lower bound on the marginal likelihood of the plug-in reverse SDE* (which by definition is exactly the generative SDE).

Recently, Durkan & Song (2021) also attempt to establish the equivalency between maximum likelihood and score matching, by showing the following relationship between the forward KL divergence and a weighted sum of score matching loss (aka the Fisher divergence):

$$D_{\mathrm{KL}}(q(y, 0)||r(y, 0)) =$$
$$\frac{1}{2}\int_0^T \mathbb{E}_{q(\cdot, s)}\left[||\nabla \log r(Y_s, s) - \nabla \log q(Y_s, s)||_{gg^\top}^2\right] ds \qquad (15)$$

where $r(y, s)$ is the density of $Y_s$ solving the same inference SDE with the initial condition $y_0 \sim r(\cdot, 0)$, assuming

---
[4]Note that $\mu$ and $\sigma$ run backward in time from $T$, whereas $a$ runs forward.

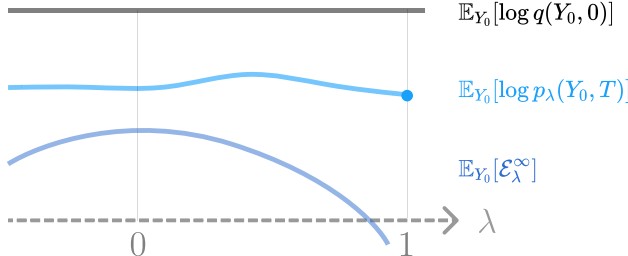

Figure 2. Lower bound on the marginal likelihood of a continuum of plug-in reverse SDEs. The lower bound is optimized when the score matching loss is minimized, which will push up the entire dark blue curve.

$q(y, T) = r(y, T)$. However, it is inaccurate to claim that score matching is equivalent to maximum likelihood. This is because if we simply let $r(y, 0) = p(y, T)$, *i.e.* the density of the generative SDE evaluated at $y$, $r(y, s)$ will not necessarily be the same as either $p(y, T - s)$ or $s_\theta(y, s)$. This means the KL divergence is not equal to the integral of the weighted score matching loss $\mathbb{E}[\frac{1}{2}||s_\theta - \nabla \log q||_{gg^\top}^2]$. In fact, the latter corresponds to a lower bound on the likelihood (the cross-entropy term of the KL) up to some constant, as equation (14) suggests.

More generally, we can apply our analysis to a family of plug-in reverse SDEs indexed by some parameter $\lambda \le 1$, which includes the plug-in reverse SDE (13) and an equivalent ODE as special cases with $\lambda = 0$ and $\lambda = 1$. In Appendix D we show that the average CT-ELBO of the $\lambda$ plug-in reverse SDE is also equivalent to the ISM loss, similarly to (14) but up to some multiplying and additive constants. The implication is that while minimizing the score matching loss, we implicitly maximize the likelihood for a continuum of plug-in reverse SDEs which include the ODE as a limiting case. See Figure 2 for illustration. This explains the good likelihood estimate reported in Song et al. (2021). In practice, we can only estimate the ELBO of the case $\lambda = 0$ since otherwise there will be some constant we do not have access to, but their gradients can all be estimated via score matching.

## 6. Conclusion

In this work, we derive a general variational framework for estimating the marginal likelihood of continuous-time diffusion models. This framework allows us to study a wide spectrum of models, including continuous-time normalizing flows and score-based generative models. Using our framework, we show that performing score matching with a particular choice of mixture weighting is equivalent to maximizing a lower bound on the marginal likelihood of a family of plug-in reverse SDEs, of which the one used in Song et al. (2021) and the equivalent ODE are special cases.

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

## A. Marginal density of diffusion models (general case)

In Section 3, we assume $\sigma$ is position-independent for simplicity. The general case of the Fokker-Planck equation can also be represented using the Feynmann-Kac formula. Following a similar conversion as in Table 2, we have

$$p(x,T) = \mathbb{E}\left[p_0(Y_T)\exp\left(\int_0^T -\nabla\cdot\mu(Y_s, T-s) + \sum_{i,j}\partial^2_{x_i,x_j}D_{ij}(Y_s, T-s)\,\mathrm{d}s\right)\,\middle|\,Y_0 = x\right]$$

where $Y_s$ solves

$$\mathrm{d}Y = -\tilde{\mu}(Y, T-s)\,\mathrm{d}s + \sigma(Y, T-s)dB'_s$$

where $\tilde{\mu}(y,s)_i := \mu_i(y,s) - 2\sum_j \partial_{x_i}D_{ij}(y,s)$.

## B. Mixture of continuous-time flows

Continuing the discussion in Remark 1, we analyze the limit of the determinant of the Jacobian of the finite approximation given the Brownian path: $x \leftarrow x + \mu(x,t)\Delta t + \sigma(t)\Delta B_i$. When the step size decreases to 0, this should converge to the Itô integral. When $\Delta t$ is small enough, under the assumption that $\mu$ is uniformly Lipschitz, the finite approximation will be invertible for all steps. Then the determinant of the Jacobian of the overall transformation is just the product of determinant of each step:

$$\prod_i \det\left(\nabla\left(x + \mu(x,t)\Delta t + \sigma(t)\Delta B_i\right)\right) = \prod_i \det\left(\mathbf{I} + \Delta t\nabla\mu\right)$$

$$= \prod_i \left(1 + \Delta t\operatorname{Tr}(\nabla\mu) + \mathcal{O}(\Delta t^2)\right)$$

$$= \exp\left(\sum_i \log\left(1 + \Delta t\nabla\cdot\mu + \mathcal{O}(\Delta t^2)\right)\right)$$

$$= \exp\left(\sum_i \Delta t\nabla\cdot\mu + \mathcal{O}(\Delta t^2)\right)$$

$$\to \exp\left(\int \nabla\cdot\mu\right) \quad \text{as } \Delta t \to 0$$

This leads to the same derivation for the instantaneous change of variable formula for continuous time flow (Chen et al., 2018), but the argument of $\mu$ will be the solution of the Itô integral, instead of the solution of the deterministic dynamics only.

## C. Equivalent SDEs

We use the following definition to formalize what we mean by equivalent SDEs

**Definition 1** (**Equivalent processes / SDEs**). *Let $Y_s$, $\tilde{Y}_s$ and $X_t$ be stochastic processes for $0 \le s, t \le T$. If $Y_s$ and $\tilde{Y}_s$ have the same distribution for all $s$, then they are said to be equivalent. If $X_t$ and $Y_{T-t}$ have the same distribution for all $t$, then we say $X_t$ is an equivalent reverse process. Two SDEs are equivalent if the processes they induce are equivalent. Two SDEs are equivalent reverse of each other if the processes they induce are equivalent reverse of one another.*

Note that when talking about the equivalency between SDEs, the dependency on an initial condition is implied.

In this section, we show how to construct a family of equivalent (reverse) SDEs. Let $Y_s$ be a diffusion process solving

$$\mathrm{d}Y = f\,\mathrm{d}s + g\,\mathrm{d}\hat{B}_s$$

We assume $g$ is position-independent and diagonal for simplicity. Let $\lambda \le 1$, We can rearrange the Fokker-Planck equation to get

$$\partial_s q = -\nabla\cdot(fq) + \frac{1}{2}g^2 : H_q = -\nabla\cdot\left(\left(f - \frac{\lambda}{2}g^2\nabla\log q\right)q\right) + \frac{1-\lambda}{2}g^2 : H_q \tag{16}$$

Now let $f_\lambda := f - \frac{\lambda}{2}g^2\nabla\log q$, and $g_\lambda := \sqrt{1-\lambda}g$. Then the SDE $dY = f_\lambda\,ds + g_\lambda\,d\hat{B}_s$ has the same Fokker Planck equation as (16), which means the SDEs defined this way form a family of equivalent SDEs [5].

Now to construct an equivalent reverse SDE, we rearrange the Fokker Planck of this new SDE,

$$\partial_s q = -\nabla\cdot(f_\lambda q) + \frac{1}{2}g_\lambda^2 : H_q = -\nabla\cdot\left(\left(f_\lambda - g_\lambda^2\nabla\log q\right)q\right) - \frac{1}{2}g_\lambda^2 : H_q \tag{17}$$

Now let $\mu_\lambda(x,t) := g_\lambda^2(x,T-t)\nabla\log q(x,T-t) - f_\lambda(x,T-t)$ and $\sigma_\lambda = g_\lambda(x,T-t)$. Then the SDE $dX = \mu_\lambda\,dt + \sigma_\lambda\,dB_t$ with the initial condition $X_0 \sim q(\cdot,T)$ is an equivalent reverse SDE, since

$$\partial_t p = -\nabla\cdot(\mu_\lambda p) + \frac{1}{2}\sigma_\lambda^2 : H_p = \nabla\cdot\left(\left(f_\lambda - g_\lambda^2\nabla\log q\right)p\right) + \frac{1}{2}g_\lambda^2 : H_p \tag{18}$$

is the time-reversal of (17). This also means there is a family of plug-in reverse SDEs parameterized by $\lambda$ and $\boldsymbol{s}_\theta$:

$$dX = (g_\lambda^2\boldsymbol{s}_\theta - f_\lambda)\,dt + \sigma_\lambda\,dB_t \tag{19}$$

$$= \left(\left(1 - \frac{\lambda}{2}\right)g^2\boldsymbol{s}_\theta - f\right)dt + \sqrt{1-\lambda}g\,dB_t \tag{20}$$

The plug-in reverse SDE used by Song et al. (2021) corresponds to $\lambda = 0$, and the equivalent (plug-in) reverse ODE corresponds to $\lambda = 1$. See Figure 3 for the simulation.

## D. Score matching and plug-in reverse SDEs

In Section 5 we establish the connection between the score matching loss and the CT-ELBO of the plug-in reverse SDE for $\lambda = 0$. If we want to do the same for different values of $\lambda$, we need to make sure the generative and inference SDEs have the same diffusion coefficient (this is to make sure the Radon-Nikodym derivative is finite). In light of this, we define the following generative and inference pair

$$dX = \left(\left(1 - \frac{\lambda}{2}\right)g^2\boldsymbol{s}_\theta - f\right)dt + \sqrt{1-\lambda}g\,dB_t \quad\text{and}\quad dY = \left(f - \frac{\lambda}{2}g^2\nabla\log q\right)ds + \sqrt{1-\lambda}g\,d\hat{B}_s \tag{21}$$

Note that this is just the same equivalent SDE and equivalent (plug-in) reverse SDE from the Appendix C. We show that maximizing the ELBO of this family of plug-in reverse SDEs is also equivalent to performing score matching.

**Theorem 2** (**Plug-in reverse SDE ELBO**). *Assume the generative and inference SDEs follow (21). For $\lambda < 1$, then the CT-ELBO (denoted by $\mathcal{E}_\lambda^\infty$) can be written as*

$$\mathcal{E}_\lambda^\infty = \mathbb{E}_{Y_T}[\log p_0(Y_T)\,|\,Y_0 = x] - \int_0^T \left(1 - \frac{\lambda}{2}\right)\mathbb{E}_{Y_s}\left[\frac{1}{2}||\boldsymbol{s}_\theta||_{g^2}^2 + \nabla\cdot\left(g^2\boldsymbol{s}_\theta - \left(\frac{2}{2-\lambda}\right)f\right)\,\Big|\,Y_0 = x\right]$$

$$+ \frac{\lambda}{2}\mathbb{E}_{Y_s}\left[\frac{1}{2}||\boldsymbol{s}_\theta||_{g^2}^2 - g^2\boldsymbol{s}_\theta^\top\nabla\log q(Y_s,s)\,\Big|\,Y_0 = x\right]$$

$$+ \frac{\lambda^2}{4(1-\lambda)}\mathbb{E}_{Y_s}\left[\frac{1}{2}||\boldsymbol{s}_\theta - \nabla\log q(Y_s,s)||_{g^2}^2\,\Big|\,Y_0 = x\right]\,ds \tag{$*$}$$

*As a result, averaging the ELBO over the data distribution and applying the identity (3) yield*

$$\mathbb{E}_{Y_0}[\mathcal{E}_\lambda^\infty] = \mathbb{E}_{Y_T}[\log p_0(Y_T)] - \int_0^T\left(1 + \frac{\lambda^2}{4(1-\lambda)}\right)\mathbb{E}_{Y_s}\left[\frac{1}{2}||\boldsymbol{s}_\theta||_{g^2}^2 + \nabla\cdot(g^2\boldsymbol{s}_\theta)\right]ds + \text{Const.} \tag{22}$$

$$= \mathbb{E}_{Y_0}[\mathcal{E}_0^\infty] - \left(\frac{\lambda^2}{4(1-\lambda)}\right)\int_0^T\mathbb{E}_{Y_s}\left[\frac{1}{2}||\boldsymbol{s}_\theta(Y_s,s) - \nabla\log q(Y_s,s)||_{g^2}^2\right]ds \tag{23}$$

Before proving this theorem, we first make a few remarks. First, setting $\lambda = 0$, this ELBO will reduce to (14). Second, (22) tells us that while matching the score, we implicitly maximize the likelihood of the entire family of plug-in reverse

---

[5]Note that more generally the same would also hold if we let $\lambda$ be a time-dependent function.

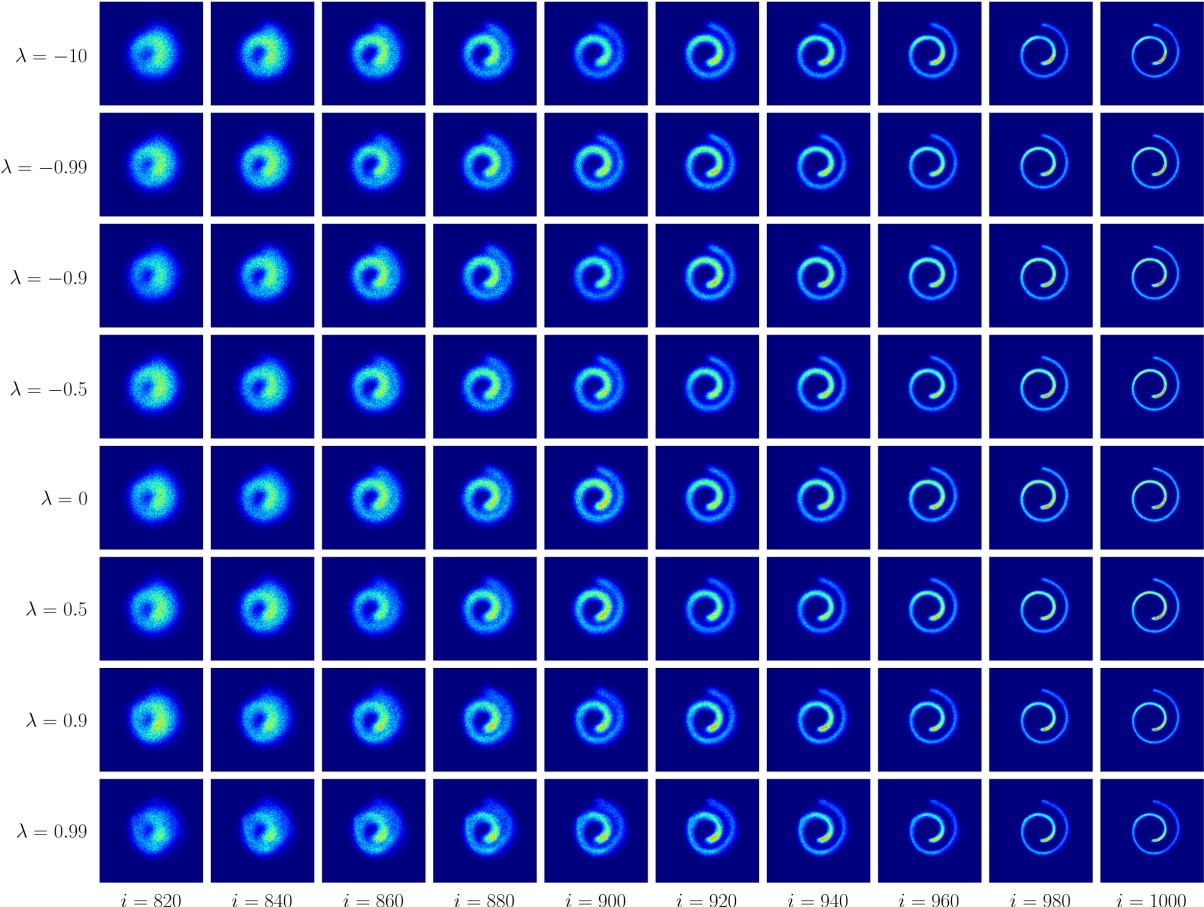

*Figure 3.* Samples from plug-in reverse SDEs with different $\lambda$ values (rows). We use the same score function $\boldsymbol{s}_\theta$ trained on the Swiss roll dataset, and plug it into (20). For generation, we use the Euler Maruyama method with a step size of $\Delta t = 1/1000$. We visualize the samples for the $i$-th iterates (columns).

SDEs. Third, (23) tells us that the average CT-ELBO is maximized when $\lambda = 0$ (recall Figure 2). Lastly, the theorem excludes the case where $\lambda = 1$, *i.e.* the equivalent ODE, since otherwise there will be a division-by-zero problem. But an ODE can be seen as having $\lambda$ very close to 1, which will make the SDE effectively deterministic in practice. This explains the low BPD of the equivalent plug-in ODE reported in Song et al. (2021).

*Proof.* Plugging (21) in (4) and (12), we get

$$\mu = \left(1 - \frac{\lambda}{2}\right)g^2 \boldsymbol{s}_\theta - f$$

$$\sigma = \sqrt{1 - \lambda}g$$

$$a = \frac{1}{\sqrt{1 - \lambda}}\left[(1 - \lambda)g\boldsymbol{s}_\theta + \frac{\lambda}{2}g\left(\boldsymbol{s}_\theta - \nabla \log q\right)\right]$$

Then we have

$$\frac{1}{2}||a||_2^2 = \frac{1}{2(1 - \lambda)}\left[(1 - \lambda)^2 ||\boldsymbol{s}_\theta||_{g^2}^2 + (1 - \lambda)\lambda g^2 \boldsymbol{s}_\theta^\top (\boldsymbol{s}_\theta - \nabla \log q) + \frac{\lambda^2}{4}||\boldsymbol{s}_\theta - \nabla \log q||_{g^2}^2\right]$$

$$= \left(1 - \frac{\lambda}{2}\right)\frac{1}{2}||\boldsymbol{s}_\theta||_{g^2}^2 + \frac{\lambda}{2}\left(\frac{1}{2}||\boldsymbol{s}_\theta||_{g^2}^2 - g^2 \boldsymbol{s}_\theta^\top \nabla \log q\right) + \frac{\lambda^2}{4(1 - \lambda)}\frac{1}{2}||\boldsymbol{s}_\theta - \nabla \log q||_{g^2}^2$$

$$\nabla \cdot \mu = \left(1 - \frac{\lambda}{2}\right)\nabla \cdot \left(g^2 \boldsymbol{s}_\theta - \left(\frac{2}{2 - \lambda}\right)f\right)$$

Summing up these two parts gives us $(*)$. Under the expectation, we can rewrite $\mathbb{E}_{Y_s}[g^2 \boldsymbol{s}_\theta^\top \nabla \log q] = -\mathbb{E}_{Y_s}[\nabla \cdot (g^2 \boldsymbol{s}_\theta)]$ using integration by parts (*i.e.* the score matching identity). $\square$

## E. Feynman-Kac formula

**Assumption 1 (Feynman-Kac).** *We assume the following: There exist some constants $B_h, B_v > 0$ and $p_h, p_v \geq 1$ such that $h \in C^0(\mathbb{R}^d)$ and $v \in C^{2,1}(\mathbb{R}^d \times [0, T])$ satisfy*

$$|h(y)| \leq B_h \left(1 + ||y||^{2p_h}\right) \text{ or } h(y) \geq 0 \tag{24}$$

$$\max_{0 \leq s \leq T} |v(y, s)| \leq B_v \left(1 + ||y||^{2p_v}\right) \tag{25}$$

**Theorem 3 (Feynman-Kac representation,** Chapter 5.7 of Karatzas & Shreve (2014)**).** *Let $T > 0$. Assume $v \in C^{2,1}(\mathbb{R}^d \times [0, T])$ solves*

$$\partial_t v + cv + b^\top \nabla v + A : H_v = 0 \tag{26}$$

*with the terminal condition $v(y, T) = h(y)$, where $A = \frac{1}{2}\eta\eta^\top$ for some matrix-valued function $\eta(y, s)$. Under Assumption 1, if $B_s'$ is a Brownian motion and $Y_s$ solves*

$$dY = b(Y, s)ds + \eta(Y, s) dB_s', \tag{27}$$

*with the initial datum $Y_s = y$, then*

$$v(y, s) = \mathbb{E}\left[h(Y_T)\exp\left(\int_s^T c(Y_{s'}, s')ds'\right)\bigg| Y_s = y\right] \tag{28}$$

To estimate the density $p(\cdot, T)$ of (7), we can apply the change of variable $p(x, t) := v(x, T - t)$ by letting the Feynman-Kac (F-K) coefficients correspond to their Fokker-Planck (F-P) counterparts according to Table 2. This way, solving (26) backward is equivalent to solving (7) forward, and we have the representation (8,9).

| F-K | F-P |
|---|---|
| $v(y,s)$ | $p(s, T-s)$ |
| $c(y,s)$ | $-\nabla \cdot \mu(y, T-s)$ |
| $b(y,s)$ | $-\mu(y, T-s)$ |
| $\eta(y,s)$ | $\sigma(T-s)$ |
| $g(y)$ | $p_0(y)$ |

*Table 2.* Feynman-Kac coefficients.

# F. Girsanov's Theorem

**Theorem 4** (**Girsanov theorem**, Theorem 8.6.3 of Øksendal (2003))**.** *Let $\hat{B}_s$ be an Itô process solving*

$$d\hat{B}_s = -a(\omega, s)\, ds + dB'_s \tag{29}$$

*for $\omega \in \Omega$, $0 \leq t \leq T$ and $\hat{B}_0 = 0$, where $a(\omega, s)$ satisfies the Novikov's condition $\mathbb{E}\left[\exp\left(\frac{1}{2}\int_0^T a^2 \, ds\right)\right] < \infty$. Then $\hat{B}_s$ is a Brownian motion wrt $\mathbb{Q}$ where*

$$\frac{d\mathbb{Q}}{d\mathbb{P}}(\omega) := \exp\left(\int_0^T a(\omega, s) \cdot dB'_s - \frac{1}{2}\int_0^T ||a(\omega, s)||_2^2 \, ds\right) \tag{30}$$

# G. Proof of Continuous-Time ELBO

*Proof.* By inverting the relationship (29), we have

$$\mathrm{d}B'_s = \mathrm{d}\hat{B}_s + a(\omega, s)\, \mathrm{d}s$$

This allows us to reparameterize $Y_s$ as

$$\mathrm{d}Y = -\mu\, \mathrm{d}s + \sigma\, \mathrm{d}B'_s = -\mu\, \mathrm{d}s + \sigma(\mathrm{d}\hat{B}_s + a\, \mathrm{d}s) = (-\mu + \sigma a)\, \mathrm{d}s + \sigma + \mathrm{d}\hat{B}_s$$

The log density can be written as

$$\begin{aligned}
\log \frac{\mathrm{d}\mathbb{P}}{\mathrm{d}\mathbb{Q}} &= -\int_0^T a \cdot \mathrm{d}B'_s + \frac{1}{2}\int_0^T ||a||^2 \, \mathrm{d}s \\
&= -\int_0^T a \cdot (\mathrm{d}\hat{B}_s + a\, \mathrm{d}s) + \frac{1}{2}\int_0^T ||a||^2 \, \mathrm{d}s \\
&= -\int_0^T a \cdot \mathrm{d}\hat{B}_s - \frac{1}{2}\int_0^T ||a||^2 \, \mathrm{d}s
\end{aligned}$$

Finally, since the first term is in expectation equal to zero (Øksendal, 2003, Theorem 3.2.1), we conclude the proof. $\qquad\square$

**Remark 2** (**Computation**). This lower bound can be numerically estimated by using any black box SDE solver, by augmenting the dynamic of $y$ with the accumulation of $||a||^2$ and $\nabla \cdot \mu$. Computing the divergence term $\nabla \cdot \mu$ directly can be expensive, but it can be efficiently estimated using the Hutchinson trace estimator (Hutchinson, 1989) along with reverse-mode automatic differentiation, similar to Grathwohl et al. (2018). As the parameters of both the generative and inference models are decoupled from the random variable $\hat{B}_s$, their gradients can be estimated via the reparameterization trick (Kingma & Welling, 2014; Rezende et al., 2014). Furthermore, backpropagation can also be computed using an adjoint method with a constant memory cost (Li et al., 2020).

**Remark 3** (**Drift $a$**). (i) In general, the drift term of the approximate posterior to the latent Brownian motion can be amortized, so that it will encode the information of individual datum $x$. (ii) The regularization $||a||^2$ ensures that $a$ is kept

close to 0, since it represents the deviation of the measure it induces (*i.e.* $\mathbb{Q}$) from the classical Wiener measure (which is a centered Gaussian measure). (iii) When the diffusion coefficient $\sigma$ is 0, the inference SDE reduces to the reverse dynamic of the generative ODE, and if $a = 0$ in this case, the lower bound is tight. (iv) There is generally no constraint on the form of $a(\omega, s)$, so one can potentially augment it with additional dimensions to have a non-Markovian inference SDE. For simplicity, unless otherwise specified, we assume the inference SDE is a diffusion model, i.e. $a = a(y, s)$.

**Remark 4** (**Variational gap**). Even though the inference SDE seemingly takes a simple form, it is sufficiently flexible in that $\sup \mathcal{E}^\infty = \log p(x, T)$, where the supremum is taken over all processes $a(\omega, s)$ that are progressively measurable. See Boué et al. (1998) for more details.

## H. Infinitely deep hierarchical VAE

In this section, we formally address the common belief that "diffusion models can be viewed as the continuous limit of hierarchical VAEs" (Tzen & Raginsky, 2019), and show that the CT-ELBO consistently extends their discrete-time counterpart. We do so by inspecting the ELBO of a hierarchical VAE defined as discretized[6] generative and inference SDEs. We assume the generative model (*i.e.* the decoder) follows the transitional distributions

$$p(x_{i+1}|x_i) = \mathcal{N}(x_{i+1}; \tilde{\mu}_i(x_i), \tilde{\sigma}_i^2) \tag{31}$$

$$\tilde{\mu}_i(x) = x + \Delta t \mu(x, i\Delta t) \qquad \tilde{\sigma}_i^2 = \Delta t \sigma^2(i\Delta t), \tag{32}$$

where $\Delta t = T/L$ is the step size and $L$ is the number of layers. For the inference model (*i.e.* the encoder), we assume

$$q(x_i|x_{i+1}) = \mathcal{N}(x_i; \hat{\mu}_{i+1}(x_{i+1}), \hat{\sigma}_{i+1}^2) \tag{33}$$

$$\hat{\mu}_i(x) = x + \Delta t(-\mu(x, i\Delta t) + \sigma(i\Delta t)a(x, T - i\Delta t)) \qquad \hat{\sigma}_i^2 = \Delta t \sigma^2(i\Delta t). \tag{34}$$

These transition kernels constitute a hierarchical variational autoencoder of $L$ stochastic layers, whose marginal likelihood can be lower bounded by

$$\log p(x_L) \geq \mathbb{E}_q \left[ \log p(x_0) + \sum_{i=0}^{L-1} \log \frac{p(x_{i+1}|x_i)}{q(x_i|x_{i+1})} \right] =: \mathcal{E}^L \tag{35}$$

which we refer to as the discrete-time ELBO (DT-ELBO). The reconstruction error of the stochastic layer can be seen as some form of finite difference approximation to differentiation, which gives rise to $\nabla \cdot \mu$ in the CT-ELBO in the infinitesimal limit (as $\Delta t$ approaches 0). The regularization of $||a||^2$ pops up when we compare the difference between $\tilde{\mu}_i$ and $\hat{\mu}_i$ using the Gaussian reparameterization to compute the reconstruction error. We formalize this idea in the following theorem.

**Theorem 5** (**Consistency**). *Assume $\mu$, $\sigma$, $\sigma^{-2}$, $a$, $||a||^2$ and their derivatives up to the fourth order are all bounded and continuous, and that $\sigma$ is non-singular. Then $\mathcal{E}^L \to \mathcal{E}^\infty$ as $L \to \infty$.*

This theorem tells us that the CT-ELBO we derive for continuous-time diffusion models is not that different from the traditional ELBO, and that maximizing the CT-ELBO can be seen as training an infinitely deep hierarchical VAE. The proof below formalizes the above intuition, using Taylor's theorem to control the polynomial approximation error, which will go to 0 as the step size $\Delta t$ vanishes when the number of layers $L$ increases to infinity.

*Proof.* By definition of the log transitional distributions

$$\log p(x_{i+1}|x_i) = -\frac{d}{2} \log 2\pi - \log \det(\tilde{\sigma}_i) - \frac{1}{2} ||x_{i+1} - \tilde{\mu}_i(x_i)||_{\tilde{\sigma}_i^{-2}}^2 \tag{36}$$

Using the definition of $\tilde{\mu}_i$, the quadratic term becomes

$$||x_{i+1} - x_i - \Delta t \mu(x_i, i\Delta t)||_{\tilde{\sigma}_i^{-2}}^2$$

---

[6] We follow the Euler-Maruyama (EM) scheme. Other discretization scheme may also work; we leave that for future work.

Due the the Gaussian reparameterization (under $q$), we can write

$$
\begin{aligned}
x_i &= \hat{\mu}_{i+1}(x_{i+1}) + \hat{\sigma}_{i+1}\epsilon \\
&= x_{i+1} + \Delta t\big(-\mu(x_{i+1}, (i+1)\Delta t) \\
&\qquad + \sigma((i+1)\Delta t)a(x_{i+1}, T-(i+1)\Delta t)) + \sqrt{\Delta t}\sigma((i+1)\Delta t)\epsilon
\end{aligned}
\tag{37}
$$

Plugging this into the quadratic term yields

$$
\begin{aligned}
||\cdots||^2 &= ||\Delta t(\mu(x_{i+1}, (i+1)\Delta t) - \mu(x_i, i\Delta t)) \\
&\quad - \Delta t\sigma((i+1)\Delta t)a(x_{i+1}, T-(i+1)\Delta t) - \sqrt{\Delta t}\sigma((i+1)\Delta t)\epsilon||^2
\end{aligned}
\tag{38}
$$

We take care of the deviation in $\mu$ first, by taking the Taylor expansion around $(x_i, i\Delta t)$:

$$
\mu(x_{i+1}, (i+1)\Delta t) = \mu(x_i, i\Delta t) + \nabla\mu(x_i, i\Delta t)^\top(x_{i+1} - x_i) + \mathcal{O}(\Delta t)
\tag{39}
$$

Note that the first order term wrt the time variable is also $\mathcal{O}(\Delta t)$, so it's absorbed into the remainder. Combining the last three identities, we have

$$
\begin{aligned}
&\frac{1}{2}||x_{i+1} - \tilde{\mu}_i(x_i)||^2_{\hat{\sigma}_i^{-2}} \\
&= \frac{1}{2}\epsilon^\top\sigma^\top(\sigma\sigma^\top)^{-1}\sigma\epsilon + \Delta t\,\epsilon^\top\sigma^\top\nabla\mu^\top(\sigma\sigma^\top)^{-1}\sigma\epsilon + \frac{1}{2}\Delta t\,a^\top\sigma^\top(\sigma\sigma^\top)^{-1}\sigma a \\
&\quad + o(\Delta t) + (\Delta t)^{1/2}\epsilon^\top\sigma^\top(\sigma\sigma^\top)^{-1}\sigma a
\end{aligned}
\tag{40}
$$

Note that we've dropped the arguments of the functions for notational convenience. All the $\sigma$s in the denominator are $\sigma(i\Delta t)$. The $o(\Delta t)$ term can be neglected since it decays fast enough even though there are $L = 1/\Delta t$ of them. The last term is 0 in expectation since $\epsilon$ is Gaussian distributed. To take care of the first term (*), we turn to the log density of the inference model.

$$
\log q(x_i | x_{i+1}) = -\frac{d}{2}\log 2\pi - \log\det(\hat{\sigma}_{i+1}) - \frac{1}{2}||x_i - \hat{\mu}_{i+1}(x_{i+1})||^2_{\hat{\sigma}_{i+1}^{-2}}
\tag{41}
$$

$$
= -\frac{d}{2}\log 2\pi - \log\det(\hat{\sigma}_{i+1}) - \frac{1}{2}||\hat{\sigma}_{i+1}\epsilon||^2_{\hat{\sigma}_{i+1}^{-2}}
\tag{42}
$$

Comparing the second term with (*), we have

$$
\frac{1}{2}\epsilon^\top\sigma_{i+1}^\top\left((\sigma_{i+1}\sigma_{i+1}^\top)^{-1} - (\sigma_i\sigma_i^\top)^{-1}\right)\sigma_{i+1}\epsilon,
\tag{43}
$$

where $\sigma_i := \sigma(i\Delta)$. Using the differential notation, in expectation, the above can be rewritten as

$$
\mathbb{E}\left[\frac{1}{2}\epsilon^\top\sigma^\top\left(\partial_t(\sigma\sigma^\top)^{-1}\right)\sigma\epsilon\right]dt = -\operatorname{tr}(\sigma^{-1}\partial_t\sigma)\,dt = -\partial_t\log\det(\sigma)\,dt,
\tag{44}
$$

where we used Hutchinson's trace identity and Jacobi's formula. Therefore, the summation of the differences will converge to $\log\det(\sigma(0)) - \log\det(\sigma(T))$. This quantity will be negated by summing up the differences between the normalizing constants for all $L$ terms, which gives us $\log\det(\sigma(T)) - \log\det(\sigma(0))$, by the telescoping cancellation.

Now we only have two terms from the quadratic function, which will converge to

$$
\epsilon^\top\sigma^\top\nabla\mu^\top\sigma^{-\top}\epsilon\,dt + \frac{1}{2}||a||^2\,dt
$$

Using the trace identity again, and the fact that trace is similarity-invariant, we see that the above quantity is equal to

$$
\left(\nabla\cdot\mu + \frac{1}{2}||a||^2\right)dt
$$

in expectation. Now summing up all the layers, we can decompose the approximate error as

$$\left| \mathbb{E}\left[ \sum \log \frac{p}{q} \right] - \mathbb{E}\left[ -\int \left( \nabla \cdot \mu + \frac{1}{2}||a||^2 \right) \right] \right| \leq \left| \mathbb{E}\left[ \sum \log \frac{p}{q} + \sum \left( \nabla \cdot \mu + \frac{1}{2}||a||^2 \right) \Delta t \right] \right|$$
$$+ \mathbb{E}\left[ \left| \sum \left( \nabla \cdot \mu + \frac{1}{2}||a||^2 \right) \Delta t - \int \left( \nabla \cdot \mu + \frac{1}{2}||a||^2 \right) \right| \right]$$

As all the approximation errors are bounded and converge to 0 as $L \to \infty$, the first term goes to 0 by the *Dominated Convergence Theorem*. The assumption on the coefficients also guarantees the convergence in mean square error (Milshtein, 1975) of the Euler Maruyama scheme, which implies the second term goes to 0. The same applies to the last step for the prior term: $x_0 \to y(T)$ in $L^2$. $\qquad\square$

## I. Non-uniform sampling for debiasing

We perform non-uniform sampling to debias the denoising score matching loss weighted by $\sigma_s^2/g^2$, as discussed in subsection J.2. We experiment with the variance-preserving SDE from Song et al. (2021) (originally from Ho et al. (2020)), whose drift and diffusion coefficients are

$$f(y, s) = -\frac{1}{2}\beta(s)y \tag{45}$$

$$g(y, s) = g(s) = \sqrt{\beta(s)} \tag{46}$$

where $\beta(s) = (\beta_{\max} - \beta_{\min})s + \beta_{\min}$, for some constants $\beta_{\max}$ and $\beta_{\min}$.

Solving the Fokker Planck of this SDE with a Dirac point mass as initial condition gives us a conditional Gaussian, whose variance is

$$\sigma_s^2 := \int_0^s g^2(s')ds' = \frac{1}{2}s^2(\beta_{\max} - \beta_{\min}) + s\beta_{\min} \tag{47}$$

Our goal is to sample from a density function proposal to $g^2/\sigma_s^2$ for most of the part. So for some small $s_\epsilon > 0$, we define the following unnormalized density

$$\tilde{q}_\epsilon(s) = \begin{cases} \frac{g^2(s_\epsilon)}{\sigma_{s_\epsilon}^2} & s \in [0, s_\epsilon) \\ \frac{g^2(s)}{\sigma_s^2} & s \in [s_\epsilon, T] \end{cases} \tag{48}$$

To simplify our notation, we let

$$\phi(s) := \log\left( \exp\left( \frac{1}{2}s^2(\beta_{\max} - \beta_{\min})s\beta_{\min} \right) - 1 \right) \tag{49}$$

$$\varphi(u) := \log\left( 1 + \exp\left( Zu + \phi(s_\epsilon) - \frac{g^2(s_\epsilon)}{\sigma_{s_\epsilon}^2}s_\epsilon \right) \right) \tag{50}$$

$$\tilde{\Phi}_\epsilon(s) := \begin{cases} \frac{g^2(s_\epsilon)}{\sigma_{s_\epsilon}^2}s & s \in [0, s_\epsilon) \\ \frac{g^2(s_\epsilon)}{\sigma_{s_\epsilon}^2}s_\epsilon + \phi(s) - \phi(s_\epsilon) & s \in [s_\epsilon, T] \end{cases} \tag{51}$$

where $\tilde{\Phi}_\epsilon$ is the cumulative function of the unnormalzied density. Evaluating it at $T$ gives us the normalizing constant $Z = \tilde{\Phi}_\epsilon(T)$, from which we obtain the CDF, $\Phi_\epsilon(s) = \frac{\tilde{\Phi}_\epsilon(s)}{Z}$, the pdf $q_\epsilon(s) = \frac{\tilde{q}_\epsilon(s)}{Z}$, and the inverse CDF that we need for sampling (using the inverse CDF transform):

$$\Phi_\epsilon^{-1}(u) = \begin{cases} Z\frac{\sigma^2(s_\epsilon)}{g^2(s_\epsilon)}u & u \in \left[0, s_\epsilon\frac{g^2(s_\epsilon)}{Z\sigma_{s_\epsilon}^2}\right) \\ \frac{1}{\beta_{\max} - \beta_{\min}}\left( -\beta_{\min} + \sqrt{\beta_{\min}^2 + 2(\beta_{\max} - \beta_{\min})\varphi(u)} \right) & u \in \left[s_\epsilon\frac{g^2(s_\epsilon)}{Z\sigma_{s_\epsilon}^2}, 1\right] \end{cases} \tag{52}$$

# J. Additional Experimental Findings

## J.1. Computational trade-off

Having a general framework for estimating the likelihood of diffusion processes allows us to compare a wide family of models, including continuous-time flows and plug-in reverse SDEs trained by score matching. We compare the two by measuring the negative ELBO throughout training to highlight their computation-estimation trade-off. We train the models on the Swiss roll toy data. For continuous-time flow, we set $\sigma = 0$, using the Hutchinson trace estimator following Grathwohl et al. (2018). The ELBO in this case is tight since $a$ will be penalized to be 0. We use the `torchdiffeq` library (Chen et al., 2018) for numerical integration for fairer comparison [7]. For plug-in reverse SDE, we train the drift network $a$ using SSM and DSM (for DSM the loss is weighted to reduce variance, which introduces some bias; see the next subsection). We use the variance-preserving inference SDE from Song et al. (2021), which allows us to sample $Y_s$ using a closed form formula, for $s$ sampled uniformly between $[0, T]$. The trained models are visualized in Figure 1, the learning curves presented in Figure 4.

From the learning curve figures, we see that neg-likelihood decreases rapidly for the continuous-time flow in terms of the number of parameter updates. But once the x-axis is normalized by runtime, the convergence speed becomes almost indistinguishable. This is because for continuous-time flows, numerical integration takes time, whereas for plug-in reverse SDEs, we train on a random time step $s$; that is, within a fixed amount of time the latter can make more parameter updates at the cost of some noise. Note that both models have constant memory cost (wrt $T$ or $L$, the number of integration steps), so a large batch size can be used to reduce variance for training.

## J.2. Bias and variance trade-off

The integral in equation (14) can be estimated by sampling $(Y_s, s)$, and using the Hutchinson trace estimator to estimate the divergence, which corresponds to implicit score matching. However, in practice the variance of this estimator is very high when the norm of the Jacobian $\nabla s_\theta$ is large. Another popular approach is to use the denoising estimator (recall the identity from (3)),

$$\mathbb{E}_{Y_s}\left[\frac{1}{2}||s_\theta(Y_s, s) - \nabla \log q(Y_s|Y_0)||^2_{gg^\top} \,\middle|\, Y_0 = x\right] \tag{53}$$

The inference SDE is typically chosen so that $Y_s$ can be easily sampled, e.g. following $\mathcal{N}(\mu_s, \sigma_s^2)$, where $\mu_s$ and $\sigma_s$ are functions of $Y_0$ and $s$. In this case, if we reparameterize $Y_s = \mu_s + \sigma_s \epsilon$ where $\epsilon \sim \mathcal{N}(0, \mathbf{I})$, then the score becomes $\nabla \log q = -\frac{\epsilon}{\sigma_s}$. Since $\sigma_s \to 0$ as $s \to 0$, this estimator normally has unbounded variance. A practical remedy for this is to multiply (53) by $\sigma_s^2/g^2$ (assuming $g$ is a scalar for simplicity), so that the target has constant magnitude on average $\mathbb{E}[\frac{1}{2}||\sigma_s s_\theta + \epsilon||^2]$, which would result in a biased gradient estimate with much smaller variance. We can debias this estimator by sampling $s \sim q(s) \propto g^2/\sigma_s^2$. This ratio, however, is usally not normalizable in practice (as it integrates to $\infty$). As an alternative, we consider the following unnormalized density $\tilde{q}_\epsilon(s) = g^2(s_\epsilon)/\sigma_{s_\epsilon}^2$ for $s \in [0, s_\epsilon]$, and $\tilde{q}_\epsilon(s) = g^2(s)/\sigma_s^2$ for $s \in [s_\epsilon, T]$. We experiment with this debiased procedure by sampling $s \sim q_\epsilon \propto \tilde{q}_\epsilon$, for $f$ and $g$ chosen to be the variance-preserving SDE.

We train the model on MNIST (LeCun et al., 1998) and CIFAR10 (Krizhevsky et al., 2009). We present the learning curves and the standard error of the estimate of the ELBO in Figure 5. The lower bound is estimated using the Hutchinson trace estimator with $s$ sampled uniformly from $[0, T]$, with the same batch size, so the only thing that will affect the dispersion is the magnitude of $\nabla s_\theta$. Since smaller values of $s$ are more likely to be sampled under $q_\epsilon$, the debiased model will see samples with less perturbation more often. On the contrary, sampling $s$ uniformly will bias the model to learn from noisier data, causing the learned score to be smoother. We also experiment with parameterizing $s_\theta$ vs parameterizing $a$. We find the latter parameterization to be helpful since the relationship $s_\theta = g^{-1}a$ has the effect of negating the multiplier $\sigma_s$ in the reweighted loss, i.e. $\mathbb{E}[\frac{1}{2}||\frac{\sigma_s}{g}a + \epsilon||^2]$. This is similar to the noise conditioning technique introduced in Song & Ermon (2020).

---

[7]Black-box SDE solvers such as `torchsde` (Li et al., 2020) might not be optimized for the deterministic case, since their stochastic adjoint method scales $\mathcal{O}(L \log L)$ in time whereas deterministic numerical solvers are usually faster. This matters for our runtime comparison.

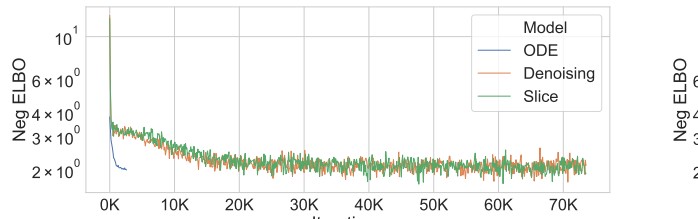 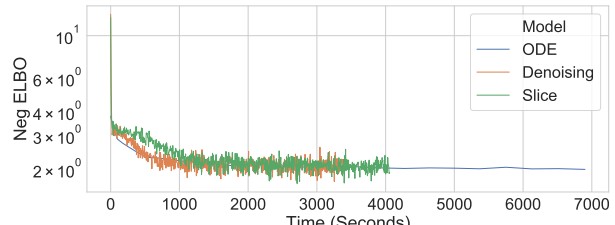

*Figure 4.* Neural ODE vs plug-in reverse SDE (denoising or slice score matching). The learning curves are presented as a function of iterations (left) and runtime (right) to emphasize the distinction between the two families of models.

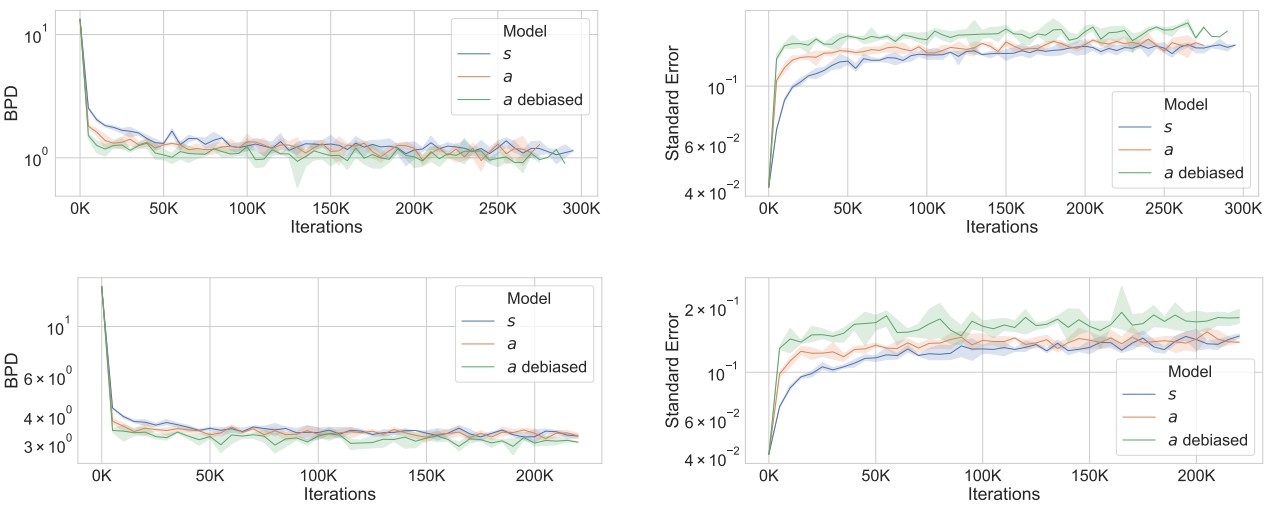

*Figure 5.* Likelihood estimation on MNIST (first row) and CIFAR10 (second row). $s$ and $a$ denotes which model we parameterize. Y-axes are bits-per-dim and the standard error of BPD of the test set. Shaded area reflects the uncertainty estimated by 3 random seeds.

# K. Experiments Details

## K.1. MNIST and CIFAR 10

We use the variance preserving SDE described in Appendix I, with $\beta_{\min} = 0.1$, $\beta_{\max} = 20$, and $T = 1$. We use the same architecture following (Ho et al., 2020) for the CIFAR10 experiment (which is a modified U-Net (Ronneberger et al., 2015)). For MNIST, we use 3 feature map resolutions (instead of 4) and reduce the number of channels from 128 to 32. Also we did not apply dropout.

For optimization, we use the Adam optimizer with a learning rate of 0.0001. We use minibatch size 128 for all experiments. We apply the standard uniform dequantization, and map the data to the real space using the logit transform (with a squeeze coefficient $\alpha = 0.05$ to avoid numerical instability). For CIFAR10, we additionally apply random horizontal flipping for regularization.

# L. Related work

**Diffusion-based generative models**   Our work lays a theoretical foundation for Song et al. (2021), which recognizes that conditional denoising score matching (Song & Ermon, 2019; 2020) and discrete-time diffusion-based generative models (Sohl-Dickstein et al., 2015; Ho et al., 2020) can be viewed as learning to revert an inference process (using the plug-in reverse SDE). This line of work has been successfully applied to modeling high dimensional natural images (Dhariwal & Nichol, 2021; Saharia et al., 2021), audio (Kong et al., 2020), 3D point cloud (Cai et al., 2020; Zhou et al., 2021), and discrete data (Hoogeboom et al., 2021).

**Score matching for energy-based models**   Besides the connection to diffusion models, score matching is also often used as a method for learning energy based models (EBM)— see Song & Kingma (2021) for a comprehensive review on useful techniques—. When used as an EBM, sampling from the conditional denoising score matching can be achieved by performing annealed importance sampling (Neal, 2001) with Langevin dynamics, which is connected to free-energy estimaton in physics (Jarzynski, 1997), wherein the *path integral* is essentially a Feynman Kac representation.

**De Bruijn's identity**   To connect maximum likelihood and score matching, Durkan & Song (2021) shows that KL divergence can be represented as an integral of weighted Fisher divergence, generalizing the case of Lyu (2009) where the inference perturbation is a simple Brownian motion. This type of formulas fall into the category of de Bruijn's identity (Cover, 1999) for relative entropy. A similar differential form result can be found in Wibisono et al. (2017).

**Learning SDEs**   Tzen & Raginsky (2019); Li et al. (2020) also propose to learn a neural SDE by applying Girsanov's theorem. The key difference is that they treat the SDE entirely as a latent variable, with an additional emission probability, whereas we use the Feynman-Kac formula to directly express the marginal density as an expectation, side-stepping the need to smooth out the density using the emission probability (which will be a Dirac point mass in our case). In their case, the inference direction is the same as the generative direction, since they infer the latent SDE directly, whereas we apply Girsanov to the Feynman-Kac diffusion (opposite the generative direction). Xu et al. (2021) further apply neural SDE as an infinitely deep Bayesian neural network.