# OpenReview forum: "A Variational Perspective on Diffusion-Based Generative Models and Score Matching"
_ICML.cc/2021/Workshop/INNF — INNF+ 2021 poster_

### Official Review · Reviewer_X1MX · 2021-06-04

**Rating:** Accept
**Confidence:** 3

**Summary:**

This works provides a theoretical underpinning of the work of Song et al. in learning the generative process of diffusion models. The main theoretical result is to show that estimating the score and using it as ''plug-in'' for the reverse SDE, corresponds to a specific setting of a variational framework, namely, maximising a lower bound on the marginal likelihood of the generative SDE.

**Justification For Rating:**

The paper is interesting and relevant for the community of this workshop. For this reason, I recommend acceptance. Perhaps, the authors could also discuss how this view can facilitate for novel algorithms.

**[Optional] Respond To Feedback Request By The Authors:**

One suggestion from my side would be to change the notation of the parametrised function that estimates the score from  $\mathbf{s}$ to something else, e.g., $r$. $s$ is also used as a time index, so it can be confusing.

---

### Official Review · Reviewer_EkgZ · 2021-06-12

**Rating:** Accept
**Confidence:** 2

**Summary:**

The paper proposes a variational interpretation of the training procedure for score-matching diffusion models. This interpretation of score-based diffusion models appears to be novel and insightful, so I recommend an accept.

I am not an expert on score-based models so my confidence score  is fairly low.

**Justification For Rating:**

The paper provides an interpretation for the training procedure of score-based models. Basically, the authors show that you can interpret the score matching objective in combination with the plug-in reverse SDE as variational inference similar to a VAE model: the latent variable  corresponds to the realization of the diffusion process. Minimizing the ELBO corresponds to optimizing the score-matching loss.

The derivation is quite non-trivial technically, and to the best of my knowledge novel. The connection between the score-matching and variational inference is also quite insightful: it provides a needed interpretation of the full procedure. So, I recommend an accept.

**[Optional] Respond To Feedback Request By The Authors:**

The writing is generally good. For a longer version of the paper, I would recommend to include a more detailed background section.

I was a bit confused by what Y_s was.

In your final ELBO in eq. (14) the integral term appears to be the score-matching objective, but what about the first term?

---

### Decision · Program_Chairs · 2021-06-14

Accept (poster)